rsos.royalsocietypublishing.org

microbiology

*Cordyceps cicadae* Shing, artificial culture, de novo transcriptome, cordycepin, RNA-Seq

**Author for correspondence:**
Xinyuan Shi
e-mail: xyshi@126.com

# Identification of cordycepin biosynthesis-related genes through de novo transcriptome assembly and analysis in *Cordyceps cicadae*

Tengfei Liu[1], Ziyao Liu[1], Xueyan Yao[2], Ying Huang[1], Qingsong Qu[1], Xiaosa Shi[1], Hongmei Zhang[1] and Xinyuan Shi[1]

[1]School of Chinese Materia Medica, Beijing University of Chinese Medicine, Beijing 100102, People's Republic of China
[2]Laboratory of Food Enzyme Engineering, Institute of Food Science and Technology, Chinese Academy of Agricultural Sciences, Beijing 100110, People's Republic of China

TL, 0000-0002-0886-8117; XS, 0000-0002-8584-1479

*Cordyceps cicadae* (Chanhua) is a parasitic fungus that grows on *Cicada flammata* larvae and is used to relieve exhaustion and treat numerous diseases, in part through its active constituent, cordycepin. We used de novo Illumina HiSeq 4000 sequencing to obtain transcriptomes of *C. cicadae* mycelium, fruiting body, and sclerotium, and identify differentially expressed genes. In the mycelium versus sclerotium libraries, 1576 upregulated and 2300 downregulated genes were identified. In the mycelium versus fruiting body and fruiting body versus sclerotium body libraries, 1604 and 1474 upregulated and 1365 and 1320 downregulated genes, respectively, were identified. Gene Ontology and Kyoto Encyclopedia of Genes and Genomes analyses identified 19 genes differentially expressed in mycelium versus fruiting body as related to the purine pathway, along with 28 and 16 genes differentially expressed in the mycelium versus sclerotium and fruiting body versus sclerotium groups, respectively. Gene expression of six key enzymes was validated by quantitative polymerase chain reaction. Specifically, 5′-nucleotidase (c62060g1) and adenosine deaminase (c35629g1) in purine nucleotide metabolism, which are involved in cordycepin biosynthesis, were significantly upregulated in the sclerotium group. These findings improved our understanding of genes involved in the biosynthesis of cordycepin and other characteristic secondary metabolites in *C. cicadae*.

rsos.royalsocietypublishing.org    R. Soc. open sci. 5: 181247

**Figure 1.** The analogues of cordycepin and the putative biosynthetic pathways of cordycepin: (*a*) chemical structure of cordycepin analogues. (*b*) AMPD, AMP deaminase; ADEK, adenylate kinase; ADK, adenosine kinase; RNR, ribonucleotide reductase; PNP, purine nucleoside phosphorylase; NT5E, 5′-nucleotidase;Cns1, oxidoreductase/dehydrogenase; Cns2, metal-dependent phosphohydrolase; Cns3/NK, N-terminal: nucleoside/nucleotide kinase (NK); C-terminal: HisG domain for ATP. The straight lines show that synthetic pathways have been validated; the dashed lines show that synthetic pathways were predicted.

## 1. Introduction

*Cordyceps cicadae* Shing (Chanhua), a parasitic fungus that can parasitize many hosts and belongs to family *Cordycipitaceae* and genus *Cordyceps*, is a source of a rare traditional Chinese medicine that has been used as an exhaustion remedy and for the treatment for numerous diseases. *Cordyceps cicadae* metabolic components include a variety of medicinal bioactive components, such as adenosine, ergosterol, cordycepin, cordycepic acids, polysaccharide, macrolides and other metabolites [1]. Recent studies have demonstrated that *C. cicadae* plays an important role in enhancing immunoregulation [2] and renal function [3], and exhibits anti-diabetic [4], anti-oxidant, antibacterial [5] and anti-tumorigenic properties [6]. In natural conditions, the fruiting bodies of *C. cicadae* form uniquely on pupated cicadae [7]; however, as rearing live cicadae is difficult, the pupae of Chinese tussah silkworm (*Antheraea pernyi*) are used for inoculation with *C. cicadae* conidia for artificial cultivation [8]. *Cordyceps cicadae* is often regarded as a substitute for *Ophiocordyceps sinensis* (DongChongXiaCao), and the medicinal value of the two species is similar. *Cordycepscicadae* is less expensive and more widely distributed than *O. sinensis*, and artificially cultivated *C. cicadae* mycelium and fruiting bodies have emerged to replace natural *C. cicadae* [9].

Cordycepin (3′-deoxyadenosine, an adenosine analogue similar to deoxyadenosine (figure 1*a*), constitutes a major bioactive nucleoside antibiotic compound found in *Cordyceps* species that was first isolated from *Cordyceps militaris* in 1950 [10]. Cordycepin possesses a wide range of pharmacological activities, such as anti-tumour, pro-immunity, antibacterial and anti-inflammatory effects [11,12]; the anti-tumour activities of this compound have been noted in squamous cell carcinoma, melanoma and cervical cancer cell proliferation [13,14]. In recent years, cordycepin has been demonstrated to induce apoptosis of lung cancer cells [15], supporting its potential role as a therapeutic agent for lung cancer and in other clinical applications. The production of medicinal cordycepin relies largely on *C. militaris* liquid fermentation, because chemical synthesis has many disadvantages, such as low yields and high

costs. Alternatively, progress in synthetic biology has enabled researchers to engineer microbes for efficient synthesis of natural compounds, such as artemisinic acid and terpenoids [16,17]. Towards this end, whole-genome sequencing of *C. militaris* was completed in 2011, which revealed that the biosynthesis of cordycepin involves formation of 2′-deoxyadenosine through a reductive mechanism [18,19].

In recent years, a series of transcriptome studies have investigated *O. sinensis* and *C. militaris*, demonstrating that biosynthetic pathways of cordycepin, as shown in figure 1*b*, involve 5′-nucleotidase, ribonucleotide and adenosine synthase as key enzymes. These studies indicated that it is possible to improve cordycepin production through analysis and regulation of its biosynthetic pathways [20–25]. Moreover, a recent study of *C. militaris* [26] revealed that the conversion of 3′-AMP to 2′-C-3′-dA is catalysed by a metal-dependent phosphohydrolase with a conserved HD motif, and 2′-C-3′-dA is further converted to cordycepin by oxidoreductase/dehydrogenase through a reduction reaction. However, the key enzymes that convert other precursor substances to cordycepin still remain unknown. By comparison of the major bioactive compounds of artificially cultured *C. militaris*, researchers have found that cordycepin levels are the highest in the fruiting bodies of pupae compared to those in other samples and are the lowest in the sclerotium [27]. Therefore, based on the similarities among *C. cicadae* developmental processes and active medicinal components (e.g. cordycepin) and those in other *Cordyceps* species, *C. militaris* and *O. sinensis* may serve as good references for gene function discovery in *C. cicadae*.

De novo RNA sequencing (RNA-Seq) constitutes a high-throughput approach that provides a new and effective method for determining transcript abundance [28] and identifying novel gene expression profiles in large-scale transcriptome studies. Moreover, direct analysis of the transcriptome of non-model organisms may be possible because this approach does not require prior knowledge of transcript sequences [29]. For example, in a previous study [30], researchers analysed key genes involved in monacolin K biosynthesis of *Monascus purpureus* by de novo RNA-Seq, identifying nine genes as being potentially associated with this process. The molecular mechanisms underlying the responses to *Beauveria bassiana* infection in silkworm (*Bombyx mori*) were also investigated via transcriptome analysis [31,32] and the drug metabolism pathways involved in toxin detoxification were found to be related to *B. bassiana* resistance and susceptibility. Furthermore, numerous comparative transcriptomic studies in plants have compared developmental stages in a single species across different tissues [33]. For *C. cicadae*, recent research has revealed the genetic basis of fungal development; however, comparative transcriptomic studies related to cordycepin biosynthesis have not yet been reported. Therefore, in this study, we investigated the possible key genes involved in the cordycepin biosynthesis pathway in the mycelium, fruiting body and sclerotium of artificially cultured *C. cicadae* using the Illumina HiSeq 4000 platform.

# 2. Material and methods

## 2.1. Isolation of fungi and sample preparation

Wild *C. cicadae* were collected from Mao Mountain in Jiangsu Province. All samples were placed in sterilized plastic bags and transported at 4°C for subsequent fungal isolation. Whole bodies of *C. cicadae* (including the fruiting bodies, stromata and sclerotia) were sterilized using 75% ethanol for 2 min and subsequently rinsed with 2.5% sodium hypochlorite for 20 min. The samples were washed five times with sterile water, and 1 g of each sample (crushed to powder in a grinder) was placed in a sterile 50 ml centrifuge tube containing 10 ml sterile phosphate-buffered saline; $10^{-1}$ and $10^{-2}$ serial dilutions of each sample suspension were spread as 0.1 ml aliquots on plates containing potato dextrose agar (PDA), and the PDA plates were then incubated at 25°C for 14 days. Distinct colonies were picked and subcultured for DNA extraction. The ITS domain of rDNA was amplified with the universal primers *ITS1* (5′-TCC GTA GGT GAA CCT GCG G-3′) and *ITS4* (5′-TCC TCC GCT TAT TGA TAT GC-3′). The thermal cycling conditions were as follows: 5 min denaturation at 95°C; 30 cycles of 30 s at 94°C, 30 s at 54°C, and 90 s at 72°C; and a final extension of 10 min at 72°C. The polymerase chain reaction (PCR) products were analysed for quality using 0.8% agarose gels (100 V, TAE buffer), ligated into a pUCm19-T vector (TransGen Biotech, Beijing, China), and then transformed into Trans5α chemically competent cells (TransGen Biotech, Beijing, China) for DNA sequencing. The resulting gene sequences were aligned with sequence data in the National Center for Biotechnology Information (NCBI) database using the BLASTN program (accession number: MF803085). A phylogenetic tree based on the ITS region was constructed using MEGA5.1 software

rsos.royalsocietypublishing.org    R. Soc. open sci. **5**: 181247

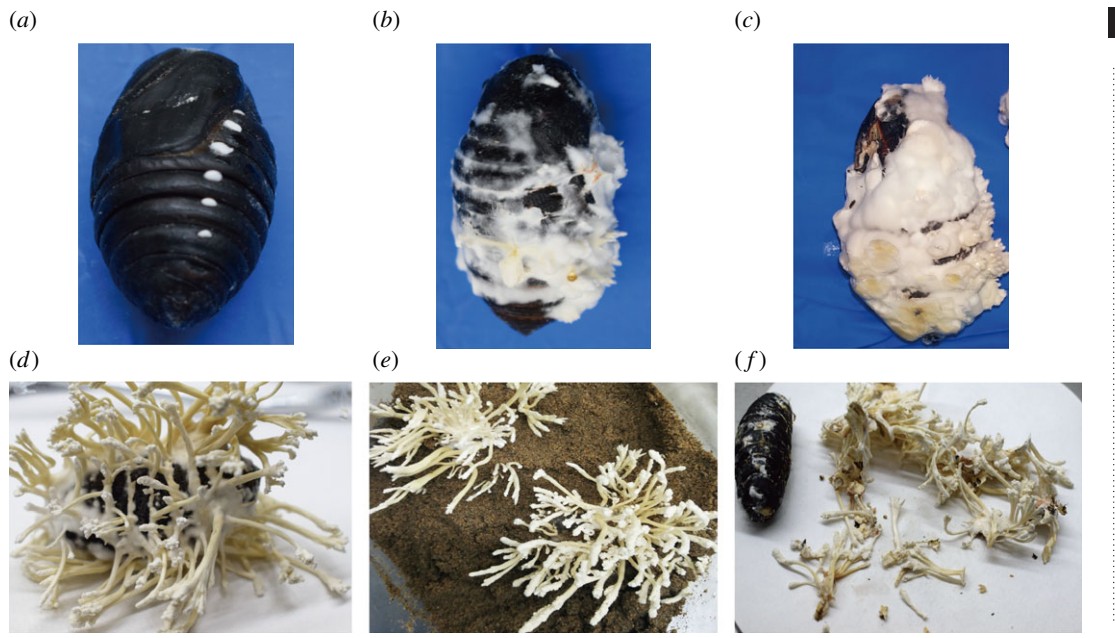

**Figure 2.** The fruiting body development of *C. cicadae* Shing: (*a*) Chinese Tussah silkmoth pupae were inoculated with conidia from the *C. cicadae* Shing strain and incubated for 7 days in the dark. (*b–d*) Infected Chinese Tussah silkmoth pupae were cultured with a 16 L : 8 D cycle at 500 lx and relative humidity greater than 80% to form sclerotium and fruiting bodies (*b*, 3 days; *c*, 6 days; *d*, 21 days). (*e*) The infected pupae were embedded in glass bottles with sterile wet soil and cultured with a 16 L : 8 D cycle at 500 lx and relative humidity greater than 80% for 21 days. (*f*) The artificially cultured *C. cicadae* were divided into the sclerotium and fruiting body.

with the neighbour-joining method [34], and the statistical analysis used bootstrapping with 1000 replications (electronic supplementary material, figure S1). The isolated strain, named *Cordyceps cicadae* strain buc, was deposited in the Guangdong Culture Collection Centre of Microbiology (GDMCC BUCM20160701).

The mycelium of *C. cicadae* (*C. cicadae* strain buc; GDMCC BUCM20160701) was inoculated onto PDA plates and incubated at 25°C for 10 days in the dark. Fresh mycelia were collected for RNA isolation, and 100 μl conidia ($1 \times 10^7$ ml$^{-1}$) of *C. cicadae* were inoculated in Chinese Tussah silkmoth pupae internally and incubated at 25°C with 80% relative humidity for 7 days in the dark (figure 2*a*). The infected pupae were then embedded in glass bottles ($10 \times 15$ cm, diameter × height) with sterile wet soil (buried in soil), and another group of infected pupae that were cultured in glass bottles without sterile wet soil (cultured in glass bottles) and allowed to form sclerotium for 10 days with a 16 L : 8 D cycle at 500 lx and a relative humidity of greater than 80%. The conditions were maintained at 25°C for 6 or 12 days to obtain developmentally mature fruiting bodies (figure 2*b,c*).

## 2.2. Cordycepin detection by high-performance liquid chromatography

The artificially cultured *C. cicadae* were divided into the sclerotium and fruiting body (figure 2*f*). These samples plus the mycelia of *C. cicadae* were air-dried at 40°C for 72 h and ground into a powder with liquid nitrogen. The amount of cordycepin was determined using a Waters e2695 system (Waters Technologies, Milford, MA, USA) equipped with four pumps, and a 2498 UV detector was used according to the method described by the Agricultural Industry Standard [35], with the following conditions: Agilent C18 chromatographic column (5 μm, $250 \times 4.6$ mm); column temperature, 25°C; flow rate, 1 ml min$^{-1}$; wavelength, 260 nm. The gradient elution conditions (flow phase A: methanol, flow phase B: water) were as follows: 0–0 min, 1% to 5% A; 10% to 15 min, 5% to 15% A; 15–20 min, 15% to 20% A; 20–30 min, 20% A; 30–35 min, 20% to 35% A; 35–40 min, 35% to 1% A.

## 2.3. RNA extraction

Total RNA was extracted from the mycelium, fruiting bodies (cultured in glass bottles) and sclerotium (buried in soil) using an RNA Prep Pure Plant Kit (TianGen, Beijing, China) according the manufacturer's instructions. Genomic DNA was removed using DNase I. The RNA quality was then

rsos.royalsocietypublishing.org    R. Soc. open sci. 5: 181247

determined with a 2100 Bioanalyzer (Agilent Technologies, Santa Clara, CA, USA) and quantified using a NanoDrop-2000 (Thermo Scientific, Wilmington, DE, USA). Only high-quality RNA ($OD_{260/280} = 1.8$–$2.2$, $OD_{260/230} \geq 2.0$, $RIN \geq 6.5$, $28S:18S \geq 1.0$, greater than 10 μg) was used to construct sequencing libraries.

## 2.4. Library preparation and illumina HiSeq 4000 sequencing

The RNA-Seq transcriptome libraries were prepared using a TruSeq RNA sample preparation kit from Illumina (San Diego, CA, USA) with 5 μg total RNA. mRNA was then isolated according to the polyA selection method using oligo(dT) beads and fragmented in fragmentation buffer. Next, double-stranded cDNA was then synthesized using a SuperScript double-stranded cDNA synthesis kit (Invitrogen, Carlsbad, CA, USA) with random hexamer primers (Illumina). The synthesized cDNA was subsequently subjected to end-repair, phosphorylation and 'A' base addition according to Illumina's library construction protocol. Libraries were size-selected for cDNA target fragments of 200–300 bp on 2% Low Range Ultra Agarose (Bio-Rad, Hercules, CA, USA) followed by PCR amplification using Phusion DNA polymerase (New England Biolabs, Ipswich, MA, USA) for 15 PCR cycles. After quantification using a TBS380 Fluorometer (Turner Biosystems Inc., Sunnyvale, CA, USA), a paired-end RNA-Seq library was sequenced with an Illumina HiSeq 4000 instrument ($2 \times 150$ bp read length). The sequencing data were deposited in the NCBI/SRA database (Bioproject: PRJNA422178; BioSample: SAMN08166800; Sequence Read Archive Database under accession number SRP126885).

## 2.5. De novo assembly and annotation

The raw paired-end reads were trimmed according to quality and length thresholds using SeqPrep (https://github.com/jstjohn/SeqPrep) and Sickle (https://github.com/najoshi/sickle) with default parameters. Clean data from the samples were then de novo assembled using Trinity V.2.4.0 (http://trinityrnaseq.sourceforge.net) [36]. All assembled transcripts were searched against the NCBI protein non-redundant (NR), String and Kyoto Encyclopedia of Genes and Genomes (KEGG) (http://www.genome.jp/kegg/) databases using BLASTX to identify the proteins that had the highest sequence similarity to the given transcripts to retrieve their functional annotations and a typical cut-off E-value of less than $1.0 \times 10^{-5}$. Blast2GO [37] (http://www.blast2go.com/b2ghome) was used to obtain gene ontology (GO) annotations of unique assembled transcripts to describe biological processes, molecular functions and cellular components. Metabolic pathway analysis was performed using the KEGG database [38].

## 2.6. Differential expression analysis and functional enrichment

To identify differentially expressed genes (DEGs) between two different samples, the expression level of each transcript was calculated according to the fragments per kilobase of exon per million mapped reads method (FPKM). RSEM (http://deweylab.biostat.wisc.edu/rsem/) [39] was used to quantify gene abundances. R statistical package software EdgeR (Empirical analysis of Digital Gene Expression; http://www.bioconductor.org/packages/2.12/bioc/html/edgeR.html) [40] was used for differential expression analysis. Thousands of independent statistical hypothesis tests were separately conducted on DEGs. Then, a $p$-value was obtained, which was corrected using the false discovery rate (FDR) method. Parameters for classifying statistically significant DEGs were as follows: at least two-fold difference in transcript abundance ($|log2FC| \geq 1$, FC: fold change in expression) and FDR less than 0.05. In addition, functional enrichment analyses, including GO and KEGG database analyses, were performed to identify the GO terms and metabolic pathways in which the DEGs were significantly enriched at a Bonferroni-corrected $p$-value of less than or equal to 0.05 compared with the whole-transcriptome background. GO functional enrichment and KEGG pathway analyses were carried out using Goatools [41] (https://github.com/tanghaibao/Goatools) and KOBAS [42,43] (http://kobas.cbi.pku.edu.cn).

## 2.7. Validation of DEGs by quantitative real-time reverse transcription (qRT)-PCR

qRT-PCR was used to confirm the data generated by high-throughput sequencing-derived gene expression profiles on an ABI PRISM 7500 detection system (Thermo Scientific, Wilmington, DE, USA) using a SYBR Real-time PCR kit (Thermo Scientific, Wilmington, DE, USA). Primer sequences are

**Table 1.** Overview of transcriptome sequencing and de novo assembly results.

| sample | total raw reads | total clean reads | error% | Q20% | Q30% | GC% |
| --- | --- | --- | --- | --- | --- | --- |
| mycelium-1 | 60 832 378 | 58 901 714 | 0.0129 | 98.06 | 94.14 | 57.8 |
| mycelium-2 | 62 011 450 | 60 093 674 | 0.0125 | 98.27 | 94.67 | 57.34 |
| mycelium-3 | 61 663 148 | 60 272 336 | 0.0137 | 97.73 | 93.32 | 58.27 |
| fruiting body-1 | 56 281 666 | 54 483 940 | 0.013 | 98.04 | 94.1 | 56.32 |
| fruiting body-2 | 66 367 912 | 64 131 966 | 0.0131 | 98 | 93.98 | 58.44 |
| fruiting body-3 | 58 874 022 | 56 909 396 | 0.013 | 98.04 | 94.1 | 57.62 |
| sclerotium-1 | 56 891 828 | 54 918 240 | 0.0132 | 97.94 | 93.84 | 58.86 |
| sclerotium-2 | 65 465 056 | 63 381 538 | 0.0129 | 98.05 | 94.11 | 58.88 |
| sclerotium-3 | 56 865 250 | 54 931 426 | 0.0131 | 97.99 | 93.96 | 58.85 |
| total | 545 252 710 | 528 024 230 | | | | |

reported in electronic supplementary material, table S2. In this study, tissue-specific expression patterns of six representative genes encoding candidate synthases (according to figure 1, the putative biosynthetic pathways of cordycepin and candidate synthases involved in cordycepin biosynthesis in previous studies [21–24]) of the purine metabolic pathway associated with mycelium, fruiting bodies and sclerotium were determined. qRT-PCR was carried out using gene-specific primers designed using Oligo 7 (Molecular Biology Insight Inc., CA, USA). PCR was performed as follows: 3 min at 94°C, followed by 35 cycles of 94°C for 30 s, 62°C for 30 s, and 72°C for 60 s. PCR amplification was monitored using 2% agarose gel electrophoresis. 18S rDNA was used as a reference gene to calculate relative expression levels based on the $2^{-\triangle\triangle CT}$ method. Statistical analysis was performed using SPSS 20.0 software (IBM SPSS Inc., Armonk, NY, USA) with an ANOVA test. Significant differences were determined using Duncan's multiple range test at the $p < 0.05$ level of significance.

# 3. Results

## 3.1. Illumina sequencing and de novo assembly

In this study, nine cDNA libraries were built from the total RNA of fresh mycelia, fruiting bodies and sclerotium, including three biological replicates; samples were designated as mycelium-1–3, fruiting body-1–3 and sclerotium-1–3. The libraries were sequenced; the number of reads per biological replicate is shown in table 1. We generated 545 252 710 raw reads and 528 024 230 clean reads. The original raw sequencing reads were purified by trimming adapters, removing reads containing poly-N and the low-quality reads containing greater than 20% of bases with a quality score less than or equal to 10, and rejecting the low-quality data (quality value less than or equal to 10 or unknown nucleotides larger than 5%) to obtain clean reads for subsequent analysis. The average Q20 and Q30 scores for clean reads among all nine samples were 98.27% and 93.32%, respectively. The clean reads were assembled using Trinity software (Trinity Technologies, CA, USA) into 70 097 unigenes and 80 403 transcripts, with the average length of the assembled unigenes being 717 bp (N50 = 1585 bp). The 80 403 transcripts comprised 46 896 transcripts (58.33%) within the length range of 1–400 bp, 9369 (11.65%) within 401–600 bp, 4737 (5.89%) within 601–800 bp, 3028 (3.77%) within 801–1000 bp, 2100 (2.61%) within 1001–1200 bp and 1695 transcripts (2.11%) within the length range of 1201–1400 bp (electronic supplementary material, figure S2A). The 70 097 unigenes comprised 45 063 unigenes (64.29%) within the length range of 1–400 bp, 8301 (11.84%) within 401–600 bp, 3832 (5.89%) within 601–800 bp, 2295 (3.27%) within 801–1000 bp, 1535 (2.19%) within 1001–1200 bp, and 1183 unigenes (1.69%) within the length range of 1201–1400 bp (electronic supplementary material, figure S2B).

To perform functional annotation of the mycelium, fruiting body and sclerotium contigs, we used BLASTX with a cut-off E-value of $1.0 \times 10^{-5}$ to search public databases, such as the Pfam, String, KEGG, SwissProt and NR databases. These additional BLAST searches revealed 15 447, 35 421, 10 437, 24 094 and 25 505 unigenes with matches in the Pfam, String, KEGG, SwissProt and NR databases, respectively. Unigene sequences showed significant similarity to known proteins in publicly available

rsos.royalsocietypublishing.org    R. Soc. open sci. **5**: 181247

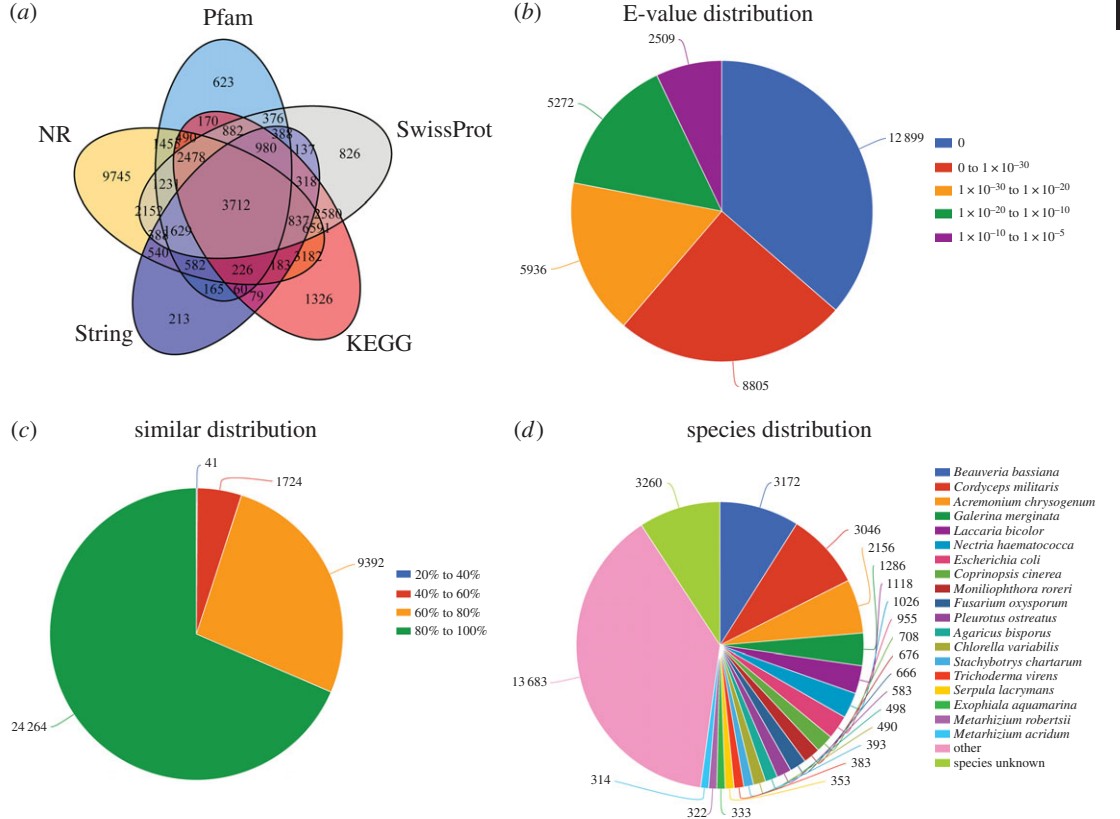

**Figure 3.** Distribution of the homology search against the NCBI database: (*a*) Venn diagram of the number of orthologous unigenes. (*b*) Distribution of E-value. (*c*) Similarity of expressed sequence tags against the NR database. (*d*) Distribution of annotated species.

databases (figure 3*a*) The E-value of 12 899 unigenes was 0, which showed that the BLASTX matches had high reliability (figure 3*b*) and a typical cut-off E-value of less than $1.0 \times 10^{-5}$. With regard to the identity distributions of the predicted proteins, most hits (68.5%) exhibited 80–100% identity with other fungi in the NR database, whereas 0.11% of the sequences had 20–40% identity (figure 3*c*). The identity and species distributions were also analysed (figure 3*d*). The species distribution of the top BLASTX hits against the NR database for the *C. cicadae* transcriptome showed that *C. cicadae* genes had the greatest number of matches with genes of *B. bassiana* and *C. militaris*.

## 3.2. Functional and GO annotations of DEGs

To explore the function of unigenes identified in the *C. cicadae* transcriptome analysis, we performed GO enrichment analysis. A total of 70 079 unigenes were categorized into 52 functional subgroups belonging to three main GO groups; i.e. cellular components, molecular functions and biological processes (figure 4). We found that the most frequent GO terms in the groups were 'metabolic process' (14 951 unigenes), 'cellular process' (12 589 unigenes) and 'catalytic activity' (12 307 unigenes).

The DEGs identified in our study were further used for GO enrichment analysis. Notably, genes upregulated during the developmental stages of *C. cicadae* were mainly involved in 'binding', 'catalytic activity', 'cellular process', 'single-organism process' and 'metabolic process'. These upregulated and downregulated genes in comparisons of mycelium (control) versus fruiting body (MF), mycelium (control) versus sclerotium (MS) and fruiting body (control) versus sclerotium (FS) were assigned to biological processes, molecular functions and cellular components using Blast2GO (electronic supplementary material, figures S5–S7). The results showed that these genes mainly contributed to 'cellular process', 'metabolic process', 'regulation of biological', 'membrane', 'binding', 'single-organism process', 'transmembrane transport' and 'catalytic activity' (electronic supplementary material, The annotation analysis results of DEGs). These up- and down-regulated genes, therefore, represent promising targets for future functional studies. Concomitant with the structural maturation occurring during the progression from mycelium to fruiting bodies (figure 2), the genes for 'nucleic acid-binding transcription factor activity', 'cell killing and transporter activity', 'immune system process' and 'biological adhesion' were

**Figure 4.** GO and KEGG analysis. GO classification of the *C. cicadae* unigene library. Histogram showing the classification of eukaryotic orthologous groups (KEGG) of the annotated unigenes.

upregulated, indicating that to support growth, the fungus must evade the recognition and defence systems of silkworms and absorb and use the nutrition of the host for multiplication.

## 3.3. Mapping and enrichment of KEGG pathways

To obtain a better understanding of the biological functions of unigenes, the annotated sequences were searched against the KEGG database. Among the 70 097 total unigenes, 33 265 (47.45%) were assigned to five main categories. The largest group comprised genes responsible for metabolism (19 125, 57.49%), including unigenes for 'global and overview maps' (6931), 'carbohydrate metabolism' (2576), 'translation' (2539), 'amino acid metabolism' (2362) and 'energy metabolism' (1778), followed by the groups comprising genes responsible for 'genetic information processing' (5234, 15.73%), 'organismal systems' (3421, 10.28%), 'cellular processes' (2769, 8.32%) and 'environmental information processing' (2716, 8.16%; electronic supplementary material, figure S3A). The KEGG enrichment analysis results are provided in electronic supplementary material, Unigenes-KEGG-table. The most significantly enriched gene in KEGG pathways was mitogen-activated protein kinase 1/3 (MAPK1_3); the mapping results are provided in electronic supplementary material, Unigene-mapping result). These annotations provided important clues for further studies of the specific development, function and pathways of *C. cicadae*. The top 10 metabolic pathways were as follows (electronic supplementary material, figure S3B): 'carbon metabolism' (1203), 'biosynthesis of amino acids' (1102), 'ribosome' (1041), 'oxidative phosphorylation' (726), 'purine metabolism' (649), 'protein processing in the endoplasmic reticulum' (617), 'Huntington's disease' (569), 'RNA transport' (544), 'spliceosome' (519) and 'Alzheimer's disease' (506). Thus, genes involved in carbohydrate metabolism and biosynthesis of secondary metabolites appeared to play significant roles in *C. cicadae*. Notably, polysaccharides represent the primary high activity components of *C. cicadae*, with those isolated from *C. cicadae* displaying potent pharmacological properties such as immunomodulatory, renoprotective and anti-oxidative properties [43,44]. In particular, the biologically active biomolecule *Cordyceps* polysaccharide is composed of simple sugars in different ratios, including mannose, glucose and fructose. Thus, the transcriptional research of carbohydrate biosynthetic pathways and the analysis of biosynthetic genes may facilitate the enhancement of *Cordyceps* polysaccharide production.

## 3.4. DEGs in the *C. cicadae* libraries

Approximately 89–92% of the total reads were successfully mapped to the *C. cicadae* transcriptome (electronic supplementary material, table S1). We then examined the number of DEGs for each of the three comparisons; i.e. MF, MS and FS, with a log2 of the fold change of 1 or more as the threshold of

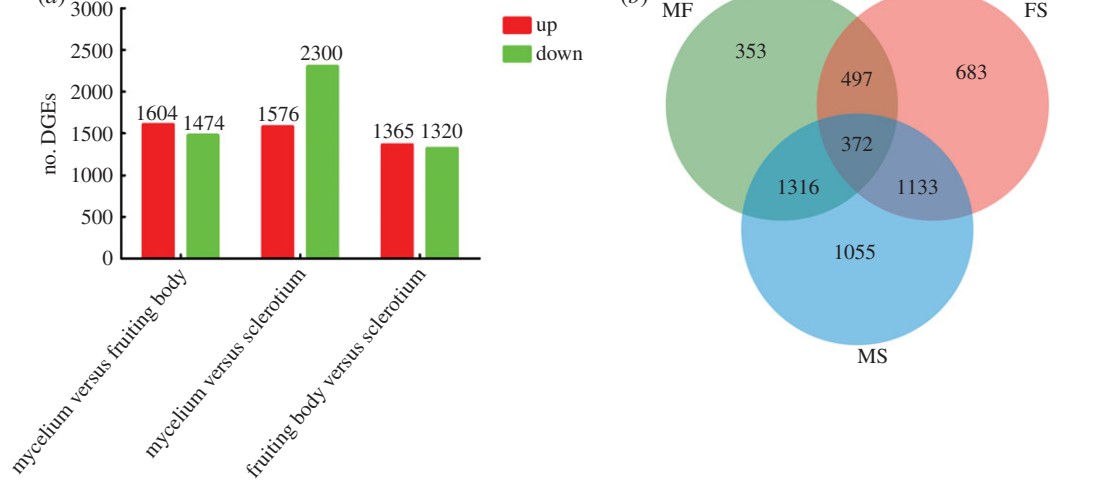

**Figure 5.** Gene expression profiles in the mycelium, fruiting body and sclerotium of artificially cultured *Cordyceps cicadae* Shing. (*a*) The numbers of up- and down-regulated genes are shown in pairwise comparisons of the three libraries. (*b*) Venn diagram of the number of DEGs based on three comparisons of mycelium versus fruiting body (MF), fruiting body versus sclerotium (FS) and mycelium versus sclerotium (MS).

**Table 2.** Cordycepin contents in the fruiting body and sclerotium.

| sample | fruiting body ($\mu$g g$^{-1}$) | sclerotium ($\mu$g g$^{-1}$) |
| --- | --- | --- |
| artificially cultured *Cordyceps cicadae* Shing | 36.69 | 83.46 |
| artificially cultured *Cordyceps cicadae* Shing (soil) | 154.53 | 319.17 |

expression fold change and an FDR $\leq 0.05$ (electronic supplementary material, figure S4A–C). The greatest number of upregulated genes was found from the MS comparison, from which a total of 3876 DEGs were detected, with 1576 upregulated and 2300 downregulated genes. This was followed by the MF and FS groups, with 1604 and 1474 upregulated genes and 1365 and 1320 downregulated genes, respectively (figure 5*a*). The Venn diagram indicated that 372 genes were significantly differentially expressed between the three comparisons MF, MS and FS (figure 5*b*). These data indicated that the gene expression patterns of the DEGs were maintained from the mycelium to sclerotium and mature fruiting body across developmental stages.

## 3.5. Purine pathway and candidate genes related to cordycepin biosynthesis

Cordycepin content in the mycelium, fruiting bodies (29 days) and sclerotium (29 days) was next determined by high-performance liquid chromatography analysis. The cordycepin contents were highest (table 2) in the sclerotium (embedded with sterile wet soil and cultured for 29 days), followed by the fruiting bodies; cordycepin was not detected in mycelium samples. Moreover, on pupae, the cordycepin level was higher in the sclerotium (embedded with sterile wet soil and cultured for 29 days) than that in other samples (electronic supplementary material, figure S8).

Numerous metabolic pathways were identified according to annotated KEGG analysis based on the MF (277 metabolic pathways), MS (292 metabolic pathways) and FS (254 metabolic pathways) transcriptomes. Most genes related to metabolism were involved in primary metabolism, such as carbohydrate metabolism, amino acid metabolism and energy metabolism. The biosynthesis of cordycepin in *C. militaris*, which is structurally similar to 2′-deoxyadenosine, in *C. militaris*, has been shown to involve the reduction of adenosine. Moreover, the biosynthetic pathway of cordycepin also involves the purine metabolic pathway [18]. Analysis of the top 20 metabolic pathways (electronic supplementary material, figure S3B) in artificially cultured *C. cicadae* indicated that the purine metabolic pathway ranked fifth among metabolic pathways. The enrichment analysis results of DEGs in KEGG are shown in electronic supplementary material, figure S9, with the most significantly enriched pathways being biosynthesis of amino acids in MF (electronic supplementary material, figure S9A), and biosynthesis of ribosomes in MS (electronic supplementary material, figure S9B) and FS (electronic supplementary material, figure

S9C) groups. As shown in figure 1, some putative biosynthetic pathways have been proposed for cordycepin; using the purine pathway (map000230) as a reference, the cordycepin metabolic pathway of *C. cicadae* was predicted (electronic supplementary material, figure S10). In the MF group, 19 participating genes were identified, with 28 and 16 participating genes identified in the MS and FS groups, respectively. The upregulated genes in the MF group were 5′-nucleotidase (c62060_g1 (2.39)), adenosine deaminase (*CECR1*; c35629_g19), DNA-directed RNA polymerase I subunit (c39342_g6), 5-hydroxyisourate hydrolase (c39057_g1), guanine deaminase (c34316_g1) and phosphoribosylamine–glycine ligase (*ADE5*), whereas the downregulated genes in the MF group were phosphoglucomutase (*PGM*), nucleoside-diphosphate kinase (*NDK*), GMP synthase (*guaA*), sulfate adenylyltransferase and DNA polymerase III subunit gamma/tau (*DPO3G*). In the MS group, the upregulated genes were ATP adenylyltransferase (*AAP1_2*), DNA-directed RNA polymerase I subunit RPA2 (*RAP2*), 5′-nucleotidase (E3.1.3.5), *CECR1*, xanthine dehydrogenase/oxidase (*XDH*), allantoicase and *ADE5*, and the downregulated genes were *PGM*, amidophosphoribosyltransferase, *NDK*, ribonucleoside-diphosphate reductase subunit M1 (*RRM1*), ribonucleoside-diphosphate reductase subunit M2 (*RRM2*), *DPO3G*, *guaA*, guanine deaminase (*guaD*), *CECR1* and urate oxidase. Finally, the upregulated genes in the MS group were *ADE5*, inosine triphosphate pyrophosphatase, *XDH*, *CECR1*, and (S)-ureidoglycine aminohydrolase, and the downregulated genes were DNA polymerase delta subunit 1, *guaD*, 5′-nucleotidase (*surE*), 5-hydroxyisourate hydrolase (*uraH*), *RRM2*, *AAP1_2*, and DNA polymerase delta subunit 1 (*POLD1*; table 3).

## 3.6. Validation of DEGs by qRT-PCR

To validate the transcriptome analysis data and provide a better understanding of the molecular basis of the metabolic pathways involved in purine biosynthesis, we selected six genes encoding key enzymes, comprising *surE* (c19447g1), *RRM1* (c19459g1), *CECR1* (c34980g1), *ADA2* (c35629g1), E3.1.3.5 (c62060g1) and *NDK* (c76121g1), to examine differences in gene expression levels using qRT-PCR (figure 6). The results showed that except for E3.1.3.5 (c62060g1) and *NDK* (c76121g1), their expression levels were consistent with those of the transcriptome analysis (figure 6a–f). In particular, the expression of both *ADA2* (c35629g1) and E3.1.3.5 (c62060g1) was highest in the sclerotium. By contrast, in the transcriptome analysis, the expression level of E3.1.3.5 (c62060g1) was highest in the fruiting body, whereas that of *NDK* (c76121g1) differed little between the fruiting body and sclerotium. These results revealed that *ADA2* (c35629g1), 5′-nucleotidase (c62060g1), *surE* (c19447g1) and *CECR1* (c34980g1) levels were strongly positively correlated with cordycepin concentrations. Thus, these results implied that differences in *C. cicadae* gene expression patterns probably accounted for the differences in cordycepin levels.

## 4. Discussion

We isolated the fungus *C. cicadae* strain buc from wild *C. cicadae* and successfully artificially cultured *C. cicadae*. The current study provides the first report of comparative de novo transcriptomic data of artificially cultured *C. cicadae*. We obtained an average of 59 755 908, 58 508 434 and 57 743 734 clean reads in the mycelium, fruiting body and sclerotium groups, respectively. Additionally, a substantial number of unigenes showed significant similarity to known proteins in publicly available Pfam, String, KEGG, SwissProt and NR databases, and the unigene lengths obtained in this study were longer than those of *Cordyceps* species previously published [22,24,45,46]. These data indicated that the transcriptome sequencing data were well assembled and the unigenes were well annotated. The transcriptome analysis of artificially cultured *C. cicadae* and the identification of related genes revealed that genes encoding 5′-nucleotidase, adenosine deaminase and nucleoside-diphosphate kinase and genes related to the glycometabolism pathway play important roles in the immunoregulation of *C. cicadae*.

Moreover, this study is the first to report predicted key enzymes for cordycepin biosynthesis in the mycelium, fruiting body and sclerotium of artificially cultured *C. cicadae*. At the fruiting body and sclerotium developmental stages, 19 and 28 genes involved in purine metabolic pathways were identified, of which 10 genes (CECR1, surE, RRM1, APA1_2, RPC1, purF, RRM2, guaA, allantoicase and XDH) were differentially expressed in the sclerotium group (buried in wet soil), whereas CECR1 (adenosine deaminase) and surE (5′-nucleotidase) appeared to closely correlate with varying cordycepin content in the mycelium, fruiting bodies and sclerotium. This indicated that the expression of enzymes was positively associated with the distribution of cordycepin content. By comparative

**Table 3.** Candidate genes enriched in the purine biosynthesis pathway.

| gene | query | gene name | E.C. number | DEGs in MS (logFC) | DEGs in MF (logFC) | DEGs in FS (logFC) |
|---|---|---|---|---|---|---|
| surE | c19447g1 | 5′-nucleotidase | 3.1.3.5 | −1.32 (1/11)[a] | n.s.[b] | −1.49 (1/8) |
| RRM1 | c19459g1 | ribonucleoside-diphosphate reductase subunit M1 | 1.17.4.1 | −2.01 (2/11) | n.s. | n.s. |
| CECR1 | c34980g1 | adenosine deaminase (CECR1) | 3.5.4.4 | −2.61 (3/11) | n.s. | −1.76 (2/8) |
| ADA2 | c35629g1 | adenosine deaminase | 3.5.4.4 | +3.03 (1/5) | +1.71(1/5) | +1.32 (1/4) |
| E3.1.3.5 | c62060g1 | 5′-nucleotidase | 3.1.3.5 | +1.28 (2/5) | +2.39 (2/5) | −1.11 (3/8) |
| NDK | c76121g1 | nucleoside-diphosphate kinase | 2.7.4.6 | −1.19 (4/11) | −1.42(1/2) | n.s. |
| purA | c7972g1 | adenylosuccinate synthase | 6.3.4.4 | −2.71 (5/11) | −1.73(2/2) | n.s. |
| RRM2 | c22306g1 | ribonucleoside-diphosphate reductase subunit M2 | 1.17.4.1 | −2.23 (6/11) | n.s. | −1.11 (4/8) |
| guaD | c34316g1 | guanine deaminase | 3.5.4.3 | −2.18 (7/11) | +1.75(3/5) | −3.93 (5/8) |
| APA1_2 | c34897g2 | ATP adenylyltransferase | 2.7.7.53 | −1.11 (9/11) | n.s. | −1.65 (6/8) |
| XDH | c37512g1 | xanthine dehydrogenase/oxidase | 1.17.1.4/1.17.3.2 | +1.83 (3/5) | n.s. | +1.36 (2/4) |
| ADE5 | c38495g2 | phosphoribosylamine–glycine ligase | 6.3.4.13/ 6.3.3.1 | +6.65 (4/5) | +3.72(4/5) | +2.92 (3/4) |
| uraH | c39057g1 | 5-hydroxyisourate hydrolase | 3.5.2.17 | −0.36 (9/11) | +1.49 (5/5) | −1.85 (7/8) |
| ITPA | c39495g1 | inosine triphosphate pyrophosphatase | 3.6.1 | +0.23 (5/5) | n.s. | +1.17 (4/4) |
| POLD1 | c55304g1 | DNA polymerase delta subunit 1 | 2.7.7.7 | −1.52 (10/11) | n.s. | −1.73 (8/8) |
| guaA | c55469g1 | GMP synthase | 6.3.5.2 | −1.22 (11/11) | n.s. | n.s. |

[a] + upregulated, − downregulated. 1/11 stands for number of genes present/total number of upregulated or downregulated genes.
[b] Not significant.

rsos.royalsocietypublishing.org    R. Soc. open sci. 5: 181247

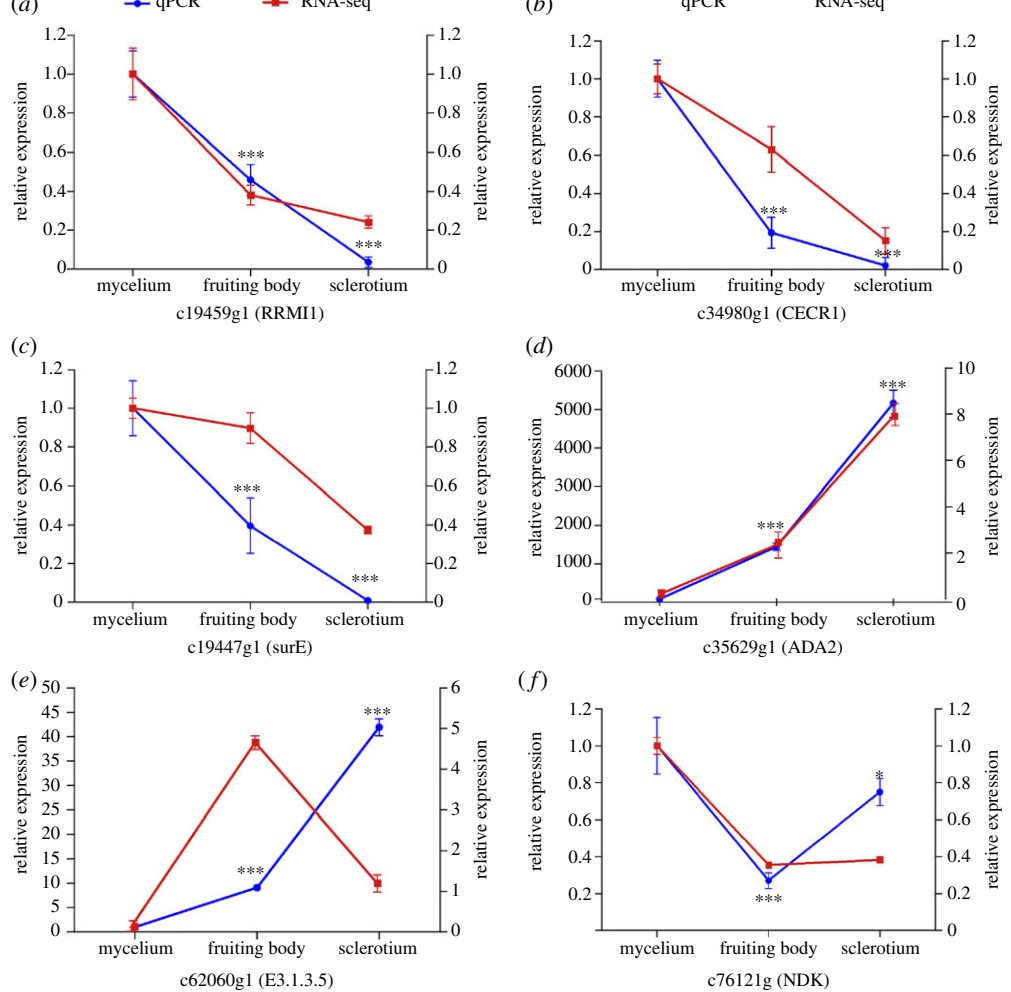

**Figure 6.** Expression profiles of six candidate genes as revealed by qPCR (left *y*-axis) and RNA-Seq (right *y*-axis) during the three developmental stages: the mean $\pm$ s.e. represents three biological replicates. Unigene IDs are marked above each figure. (*a*) c19459g1 (ribonucleoside-diphosphate reductase subunit M1). (*b*) c34980g1 (adenosine deaminase CECR1). (*c*) c19447g1 (5′-nucleotidase). (*d*) c35629g1 (adenosine deaminase ADA2). (*e*) c62060g1 (5′-nucleotidase). (*f*) c76121g1 (nucleoside-diphosphate kinase). Different numbers of asterisks represent significant differences between groups (*$p < 0.05$ ,**$p < 0.01$ ,***$p < 0.0001$).

transcriptome analysis of the mycelium, fruiting bodies and sclerotium, we found that co-expression of 5′-nucleotidase (c62060g1, upregulated; c19447g1, downregulated) and adenosine deaminase (c35629g1, upregulated; c34980g1, downregulated) was involved in the purine metabolic pathway. These findings are consistent with previous reports that deoxyadenosine, AMP, adenosine, IMP and d-AMP comprise potential precursor substances [20], and that 5′-nucleotidase constitutes an important enzyme involved in cordycepin biosynthesis [19] with potential roles in phosphorylation and dephosphorylation in the cordycepin biosynthetic pathway.

In the purine metabolism pathway, we further found some key enzymes that are probably related to cordycepin biosynthesis in different developmental stages of *C. cicadae*. Adenosine deaminase, 5′-nucleotidase, guanine deaminase, dehydrogenase/oxidase and phosphoribosylamine–glycine ligase exhibited significant association with cordycepin biosynthesis at the sclerotium developmental stage. qPCR analysis used to validate the expression levels of 5′-nucleotidase adenosine deaminase unigenes revealed that the gene for adenosine deaminase *ADA2* (c35629g1) showed the highest expression level, followed by the 5′-nucleotidase (c62060g1) gene. Although adenosine and its derivatives may be converted to cordycepin (figure 1), the subsequent steps in the cordycepin biosynthetic pathway remain unclear in *C. cicadae*. Based on our findings, a proposed biosynthetic pathway for cordycepin in *C. cicadae* is shown in figure 7. The biosynthesis of cordycepin proceeds through a reductive mechanism [18]; cloning and heterologous expression of these 5′-nucleotidases, adenosine deaminases and other key enzymes may, therefore, provide additional insight into the biosynthetic process of

rsos.royalsocietypublishing.org    R. Soc. open sci. **5**: 181247

**Figure 7.** Putative biosynthesis pathway for cordycepin in *C. cicadae*.

cordycepin. However, further research such as gene knock-out and knock-down or overexpression methods should be adopted to verify gene function.

# 5. Conclusion

In summary, we conducted the first large-scale RNA-Seq analysis of the mycelium, fruiting bodies and sclerotium of artificially cultured *C. cicadae*, by which 70 097 unigenes were generated. Our data provided comprehensive coverage of the *C. cicadae* transcriptome and revealed specific genes involved in the development of *C. cicadae*. Furthermore, we found that 5′-nucleotidase and adenosine deaminase were involved in the biosynthesis of cordycepin. Further studies are needed to elucidate the mechanism by which these enzymes contribute to cordycepin formation and the reductive modification of cordycepin biosynthesis. Our results are expected to facilitate further pharmaceutical and industrial applications of *C. cicadae* along with studies of the molecular mechanism of cordycepin biosynthesis in *C. cicadae*.

Ethics. All experimental procedures were performed according to the Regulations for the Administration of Affairs Concerning Experimental Animals (Ministry of Science and Technology, China, 2004).

Data accessibility. The full set of raw data from this study has been deposited in the National Center for Biotechnology Information's Sequence Read Archive (SRA) and is accessible through the SRA accession SRP126885 (https://www.ncbi.nlm.nih.gov/sra/SRP126885). Electronic supplementary material is available online at https://figshare.com/s/44f05ffbee9dc1be8208.

Authors' contributions. X.S. conceptualized and supervised the study. T.L., Z.L. and X.Y. completed all experiments. T.L. and Y.H. drafted and revised the manuscript. Q.Q. and X.S. performed computational analysis. H.Z. prepared experimental material. All authors gave final approval for the publication.

Competing interests. The authors declare no competing interests.

Funding. This research did not receive any specific grant from funding agencies in the public, commercial or not-for-profit sectors.

Acknowledgements. We thank the Majorbio Co. (Shanghai, China) for the Illumina deep RNA-Seq sequencing and help with data analysis. We are also grateful to two anonymous reviewers who provided comments that substantially improved the manuscript.

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
