## [Reviewer comments · Royal Society Open Science]

Review History

RSOS-181247.R0 (Original submission)

Review form: Reviewer 1

Is the manuscript scientifically sound in its present form?

No

Are the interpretations and conclusions justified by the results?

Yes

Is the language acceptable?

No

Is it clear how to access all supporting data?

Yes

Do you have any ethical concerns with this paper?

No

Have you any concerns about statistical analyses in this paper?

No

Recommendation?

Reject

Comments to the Author(s)

Major comments:

I suggest the authors organize their manuscript in a logical way, and present their data in a simple and direct way. It's better if they provide a hypothesis for their study. The language should also be polished by an English-speaking expert.

L15-24 What is the difference between "gene expression analysis" and "GO and KEGG analyses"? Each analysis produced different numbers of differential genes between libraries. Why?

L24 Why were the six genes chosen?

L66 *Ophiocordyceps sinensis* and *Hirsutella sinensis* are representing the same species. The fungus is known to produce undetectable cordycepin.

L113-117 Primers ITS1 and ITS4 are actually amplifying the ITS region instead of 18S rDNA. PCR extension time of 90 s at 72°C is too long for the 500-bp ITS region.

L155-156 Are "cultured in canned bottles" and "buried in soil" describing the same thing?

L238-250 The paragraph can be simplified to avoid duplicating methods.

L253 What's the difference between a unigene and a transcript? In later parts of the manuscript, it seems that only unigenes were analyzed, and transcripts were not mentioned again.

L311, L314, and other parts. It's not a conventional way to show and cite data by providing the file name directly in the text.

The subsection 3.5 is partially duplicating subsection 3.2.

Minor comments:

L27 are responsible

L34 Cordycipitaceae is a family name and Cordyceps is a genus name.

L47 *O. sinensis*

L105-106 What's the difference between fruiting body and stromata?

L126 "strain buc" should not be italic.

Figure 5B The venn diagram is unusual. Each big circle should be a library instead of a comparison as shown.

Review form: Reviewer 2

Is the manuscript scientifically sound in its present form?

Yes

Are the interpretations and conclusions justified by the results?

Yes

Is the language acceptable?

Yes

Is it clear how to access all supporting data?

Yes

Do you have any ethical concerns with this paper?

No

Have you any concerns about statistical analyses in this paper?

No

Recommendation?

Accept with minor revision (please list in comments)

Comments to the Author(s)

In the manuscript, the authors provide the first comparative transcriptomic data of *Cordyceps cicadae*, one of the most famous fungi as traditional Chinese medicinal material. In addition, the authors were also trying to identify genes related to cordycepin biosynthesis. This study improved our understanding of secondary metabolism in *C. cicadae*. The study is solid and the results reported are interesting. The manuscript is also nicely written. Therefore, this manuscript is worth publication. Following please a few examples to improve this manuscript.

- 1) Line 51 The second and subsequent generic name *Cordyceps* in the text should be abbreviated.
- 2) Replace *Ophiocordyceps sinensis* with *Cordyceps sinensis*.
- 3) Line 131 should be Figure 2F.
- 4) Line 331, the species name should be italicized. Please make sure all species names are italicized in this manuscript.
- 5) Line 347 should be "GO".
- 6) The text in Figures 3-5 are too small to read.
- 7) The Figures should be clearly referenced, which should be "Figure S3A", instead of "Figure S3 A". Please correct all in throughout the whole manuscript.

Review form: Reviewer 3

Is the manuscript scientifically sound in its present form?

Yes

Are the interpretations and conclusions justified by the results?

Yes

Is the language acceptable?

Yes

Is it clear how to access all supporting data?

No

Do you have any ethical concerns with this paper?

No

Have you any concerns about statistical analyses in this paper?

Yes

Recommendation?

Major revision is needed (please make suggestions in comments)

Comments to the Author(s)

This manuscript (MS) RSOS-181247 reported the "Identification of Cordycepin Biosynthesis-related genes through de novo transcriptome assembly and analysis of *Cordyceps cicadae*". The MS is relatively well written, scientifically sound and the outcomes are relatively well explained, and it is of interest to the readership of Royal Society Open Science. Therefore, I recommend its acceptance for publication after major revision.

The MS needs an extensive proofreading. There are extra full points and missing spaces between words and references. I strongly suggest reviewing all the citations, there are two changed references. I also recommend looking after the Royal Society Open Science rules for authors, example: Fig. 2 or Figure 2; Figure S1 or Fig. S1.

The table with annotation results is not available (Supplementary Materials: The annotation analysis result of DEGs.xlsx).

My comments and suggestions are in attached file (Appendix A).

Decision letter (RSOS-181247.R0)

08-Oct-2018

Dear Dr Liu,

The editors assigned to your paper ("Identification of Cordycepin Biosynthesis-related Genes Through de novo Transcriptome Assembly and Analysis of *Cordyceps cicadae*") have now received comments from reviewers. While some of the reviewers are very positive about publication, the reviewers also raise some substantive points which merit careful consideration. We would like you to revise your paper in accordance with the referee and Associate Editor suggestions which can be found below (not including confidential reports to the Editor). Please note this decision does not guarantee eventual acceptance.

Please submit a copy of your revised paper before 31-Oct-2018. Please note that the revision deadline will expire at 00.00am on this date. If we do not hear from you within this time then it will be assumed that the paper has been withdrawn. In exceptional circumstances, extensions may be possible if agreed with the Editorial Office in advance. We do not allow multiple rounds of revision so we urge you to make every effort to fully address all of the comments at this stage. If deemed necessary by the Editors, your manuscript will be sent back to one or more of the original reviewers for assessment. If the original reviewers are not available, we may invite new reviewers.

- Data accessibility

If you wish to submit your supporting data or code to Dryad (<http://datadryad.org/>), or modify your current submission to dryad, please use the following link:
<http://datadryad.org/submit?journalID=RSOS&manu=RSOS-181247>

- Competing interests

- Authors' contributions

- Acknowledgements

- Funding statement

Please note that Royal Society Open Science charge article processing charges for all new submissions that are accepted for publication. Charges will also apply to papers transferred to Royal Society Open Science from other Royal Society Publishing journals, as well as papers submitted as part of our collaboration with the Royal Society of Chemistry (<http://rsos.royalsocietypublishing.org/chemistry>). If your manuscript is newly submitted and subsequently accepted for publication, you will be asked to pay the article processing charge, unless you request a waiver and this is approved by Royal Society Publishing. You can find out more about the charges at <http://rsos.royalsocietypublishing.org/page/charges>. Should you have any queries, please contact opscience@royalsociety.org.

Kind regards,
 Royal Society Open Science Editorial Office
 Royal Society Open Science
opscience@royalsociety.org

on behalf of Dr Berat Haznedaroglu (Associate Editor) and Prof. Steve Brown (Subject Editor)
opscience@royalsociety.org

Comments to Author:

Reviewers' Comments to Author:
 Reviewer: 1

Comments to the Author(s)

Major comments:

I suggest the authors organize their manuscript in a logical way, and present their data in a simple and direct way. It's better if they provide a hypothesis for their study. The language should also be polished by an English-speaking expert.

L15-24 What is the difference between "gene expression analysis" and "GO and KEGG analyses"? Each analysis produced different numbers of differential genes between libraries. Why?

L24 Why were the six genes chosen?

L66 *Ophiocordyceps sinensis* and *Hirsutella sinensis* are representing the same species. The fungus is known to produce undetectable cordycepin.

L113-117 Primers ITS1 and ITS4 are actually amplifying the ITS region instead of 18S rDNA. PCR extension time of 90 s at 72°C is too long for the 500-bp ITS region.

L155-156 Are "cultured in canned bottles" and "buried in soil" describing the same thing?

L238-250 The paragraph can be simplified to avoid duplicating methods.

L253 What's the difference between a unigene and a transcript? In later parts of the manuscript, it seems that only unigenes were analyzed, and transcripts were not mentioned again.

L311, L314, and other parts. It's not a conventional way to show and cite data by providing the file name directly in the text.

The subsection 3.5 is partially duplicating subsection 3.2.

Minor comments:

L27 are responsible

L34 Cordycipitaceae is a family name and Cordyceps is a genus name.

L47 *O. sinensis*

L105-106 What's the difference between fruiting body and stromata?

L126 "strain buc" should not be italic.

Figure 5B The venn diagram is unusual. Each big circle should be a library instead of a comparison as shown.

Reviewer: 2

Comments to the Author(s)

In the manuscript, the authors provide the first comparative transcriptomic data of *Cordyceps cicadae*, one of the most famous fungi as traditional Chinese medicinal material. In addition, the authors were also trying to identify genes related to cordycepin biosynthesis. This study improved our understanding of secondary metabolism in *C. cicadae*. The study is solid and the results reported are interesting. The manuscript is also nicely written. Therefore, this manuscript is worth publication. Following please a few examples to improve this manuscript.

- 1) Line 51 The second and subsequent generic name *Cordyceps* in the text should be abbreviated.
- 2) Replace *Ophiocordyceps sinensis* with *Cordyceps sinensis*.
- 3) Line 131 should be Figure 2F.
- 4) Line 331, the species name should be italicized. Please make sure all species names are italicized in this manuscript.
- 5) Line 347 should be "GO".
- 6) The text in Figures 3-5 are too small to read.
- 7) The Figures should be clearly referenced, which should be "Figure S3A", instead of "Figure S3 A". Please correct all in throughout the whole manuscript.

Reviewer: 3

Comments to the Author(s)

This manuscript (MS) RSOS-181247 reported the "Identification of Cordycepin Biosynthesis-related genes through de novo transcriptome assembly and analysis of *Cordyceps cicadae*". The MS is relatively well written, scientifically sound and the outcomes are relatively well explained, and it is of interest to the readership of Royal Society Open Science. Therefore, I recommend its acceptance for publication after major revision.

The MS needs an extensive proofreading. There are extra full points and missing spaces between words and references. I strongly suggest reviewing all the citations, there are two changed references. I also recommend looking after the Royal Society Open Science rules for authors, example: Fig. 2 or Figure 2; Figure S1 or Fig. S1.

The table with annotation results is not available (Supplementary Materials: The annotation analysis result of DEGs.xlsx).

My comments and suggestions are in attached file.

Author's Response to Decision Letter for (RSOS-181247.R0)

See Appendix B.

RSOS-181247.R1 (Revision)

Review form: Reviewer 1

Is the manuscript scientifically sound in its present form?

Yes

Are the interpretations and conclusions justified by the results?

Yes

Is the language acceptable?

Yes

Is it clear how to access all supporting data?

Yes

Do you have any ethical concerns with this paper?

No

Have you any concerns about statistical analyses in this paper?

No

Recommendation?

Accept with minor revision (please list in comments)

Comments to the Author(s)

Title in Cordyceps cicadae

Introduction Cordyceps cicadae can parasitize many hosts, not just Cicada flammata.

3.2 The last sentence is unclear. Why?

3.3 Why did you use fruiting bodies cultured in canned bottles while sclerotium samples buried in soil?

4.1 Mycelium-1-3

References Not all authors were listed consistent with papers for some literatures (e.g., 8) while only one author was listed for some followed by et al (e.g., 19).

Review form: Reviewer 3

Is the manuscript scientifically sound in its present form?

Yes

Are the interpretations and conclusions justified by the results?

Yes

Is the language acceptable?

Yes

Is it clear how to access all supporting data?

Yes

Do you have any ethical concerns with this paper?

No

Have you any concerns about statistical analyses in this paper?

No

Recommendation?

Accept as is

Comments to the Author(s)

The authors answered the questions appropriately, reviewed the text formatting and the English language. I recommend its acceptance for publication.

Decision letter (RSOS-181247.R1)

14-Nov-2018

Dear Dr Liu:

On behalf of the Editors, I am pleased to inform you that your Manuscript RSOS-181247.R1 entitled "Identification of Cordycepin Biosynthesis-related Genes Through de novo Transcriptome Assembly and Analysis of *Cordyceps cicadae*" has been accepted for publication in Royal Society Open Science subject to minor revision in accordance with the referee suggestions. Please find the referees' comments at the end of this email.

The reviewers and Subject Editor have recommended publication, but also suggest some minor revisions to your manuscript. Therefore, I invite you to respond to the comments and revise your manuscript.

- **Ethics statement**

- **Data accessibility**

If you wish to submit your supporting data or code to Dryad (<http://datadryad.org/>), or modify your current submission to dryad, please use the following link:
<http://datadryad.org/submit?journalID=RSOS&manu=RSOS-181247.R1>

- **Competing interests**

- **Authors' contributions**

- **Acknowledgements**

- **Funding statement**

Because the schedule for publication is very tight, it is a condition of publication that you submit the revised version of your manuscript before 23-Nov-2018. Please note that the revision deadline will expire at 00.00am on this date. If you do not think you will be able to meet this date please let me know immediately.

Please note that Royal Society Open Science charge article processing charges for all new submissions that are accepted for publication. Charges will also apply to papers transferred to Royal Society Open Science from other Royal Society Publishing journals, as well as papers submitted as part of our collaboration with the Royal Society of Chemistry (<http://rsos.royalsocietypublishing.org/chemistry>). If your manuscript is newly submitted and subsequently accepted for publication, you will be asked to pay the article processing charge, unless you request a waiver and this is approved by Royal Society Publishing. You can find out more about the charges at <http://rsos.royalsocietypublishing.org/page/charges>. Should you have any queries, please contact openscience@royalsociety.org.

on behalf of Dr Berat Haznedaroglu (Associate Editor) and Professor Steve Brown (Subject Editor)
openscience@royalsociety.org

Reviewer comments to Author:
Reviewer: 1

Comments to the Author(s)
Title in *Cordyceps cicadae*
Introduction *Cordyceps cicadae* can parasitize many hosts, not just *Cicada flammata*.

3.2 The last sentence is unclear. Why?

3.3 Why did you use fruiting bodies cultured in canned bottles while sclerotium samples buried in soil?

4.1 Mycelium-1-3

References Not all authors were listed consistent with papers for some literatures (e.g., 8) while only one author was listed for some followed by et al (e.g., 19).

Reviewer: 3

Comments to the Author(s)

The authors answered the questions appropriately, reviewed the text formatting and the English language. I recommend its acceptance for publication.

Author's Response to Decision Letter for (RSOS-181247.R1)

See Appendix C.

Decision letter (RSOS-181247.R2)

20-Nov-2018

Dear Dr Liu,

I am pleased to inform you that your manuscript entitled "Identification of Cordycepin Biosynthesis-related Genes Through de novo Transcriptome Assembly and Analysis of *Cordyceps cicadae*" is now accepted for publication in Royal Society Open Science.

Kind regards,

Andrew Dunn

on behalf of Dr Berat Haznedaroglu (Associate Editor) and Steve Brown (Subject Editor)
openscience@royalsociety.org

Appendix A

Referee's Comments:

This manuscript (MS) RSOS-181247 reported the “Identification of Cordycepin Biosynthesis-related genes through de novo transcriptome assembly and analysis of *Cordyceps cicadae*”.

The MS is relatively well written, scientifically sound and the outcomes are relatively well explained, and it is of interest to the readership of Royal Society Open Science. Therefore, I recommend its acceptance for publication after major revision.

The MS needs an extensive proofreading. There are extra full points and missing spaces between words and references. I strongly suggest reviewing all the citations, there are two changed references. I also recommend looking after the Royal Society Open Science rules for authors, example: Fig. 2 or Figure 2; Figure S1 or Fig. S1.

The table with annotation results is not available (Supplementary Materials: The annotation analysis result of DEGs.xlsx).

Line 38: “...macrolides, and other metabolites[1].”

It should be “...macrolides, and metabolites [1].”

Lines 38 to 41: the authors described many studies of cordycepin function but did not cite the reference (ref) of each study. I recommend citing the reference for each function: immunoregulation [ref], renal function [ref], anti-fatigue [ref], anti-aging [ref], and anti-tumor properties [ref]; and promotes hemopoietic function [ref], sedation [ref], and hypnosis [ref].

Here are some reference examples for renal fibrosis, anticancer and anti-aging:

Zheng R, Zhu R, Li X, Li X, Shen L, Chen Y, Zhong Y, Deng Y. N6-(2-Hydroxyethyl) Adenosine From *Cordyceps cicadae* Ameliorates Renal Interstitial Fibrosis and Prevents Inflammation via TGF- β 1/Smad and NF- κ B Signaling Pathway. *Front Physiol.* 2018 Sep 4;9:1229. doi: 10.3389/fphys.2018.01229. eCollection 2018. PubMed PMID: 30233405; PubMed Central PMCID: PMC6131671.

Zhao X, Yu XH, Zhang GY, Zhang HY, Liu WW, Zhang CK, Sun YJ, Ling JY. Aqueous Extracts of *Cordyceps kyushuensis* Kob Induce Apoptosis to Exert Anticancer Activity. *Biomed Res Int.* 2018 Aug 9;2018:8412098. doi: 10.1155/2018/8412098. eCollection 2018. PubMed PMID: 30175146; PubMed Central PMCID: PMC6106948.

Zhang Y, Zhang XX, Yuan RY, Ren T, Shao ZY, Wang HF, Cai WL, Chen LT, Wang XA, Wang P. Cordycepin induces apoptosis in human pancreatic cancer cells via the mitochondrial-mediated intrinsic pathway and suppresses tumor growth in vivo. *Onco Targets Ther.* 2018 Aug 1;11:4479-4490. doi: 10.2147/OTT.S164670. eCollection 2018. PubMed PMID: 30122940; PubMed Central PMCID: PMC6078188.

Cheng WY, Wei XQ, Siu KC, Song AX, Wu JY. Cosmetic and Skincare Benefits of Cultivated Mycelia from the Chinese Caterpillar Mushroom, *Ophiocordyceps sinensis* (Ascomycetes). *Int J Med Mushrooms.* 2018;20(7):623-636. doi: 10.1615/IntJMedMushrooms.2018026883. PubMed PMID: 30055554.

Line 53: “...and anti-inflammatory effects[6,7]; ...”

Corrected: "...and anti-inflammatory effects [6,7]; ..."

Line 55: "...cancer cell proliferation[8,9]." It should be "...cancer cell proliferation [8,9]."

Line 55: The reference [8] is incorrect. The reference [8] is about bioinformatics "RSEM: accurate transcript quantification from RNA-Seq data ..." The correct reference is the one numbered as [32] "Anti-cancer effects of cordycepin on oral squamous cell cancer..."

I suggest reviewing the citations and change the numbers and references appropriately.

Line 56: "... lung cancer cells[10], ..."

Corrected: "... lung cancer cells [10], ..."

Line 61: "... and terpenoids[11,12]."

Corrected: "... and terpenoids [11,12]."

Line 64: "... a reductive mechanism[13,14]."

Corrected: "... a reductive mechanism [13,14]."

Line 70: "... pathways[15-20], recent research of *C. militaris*[21] ..."

Corrected: "... pathways [15-20], recent research of *C. militaris* [21] ..."

Line 77: "... sclerotium[22]."

Corrected: "... sclerotium [22]."

Line 83: "... transcript abundance[23] and ..."

Corrected: "... transcript abundance [23] and ..."

Line 83: The reference [23] is about whole transcriptome analysis of single cell, this manuscript did not use single cell technology. I suggest change the reference for a whole transcriptome analysis like the reference [30] and:

Ungaro A, Pech N, Martin JF, McCairns RJS, Mévy JP, Chappaz R, Gilles A. Challenges and advances for transcriptome assembly in non-model species. PLoS One. 2017 Sep 20;12(9):e0185020. doi:10.1371/journal.pone.0185020. eCollection 2017. PubMed PMID: 28931057; PubMed Central PMCID: PMC5607178.

Francis WR, Christianson LM, Kiko R, Powers ML, Shaner NC, Haddock SH. A comparison across non-model animals suggests an optimal sequencing depth for de novo transcriptome assembly. BMC Genomics. 2013 Mar 12;14:167. doi: 10.1186/1471-2164-14-167. PubMed PMID: 23496952; PubMed Central PMCID: PMC3655071.

Line 85: I suggest substituting "nonmodel" for non-model organisms.

Line 86 to 87: "... transcript sequences[24].In a previous study[25], ..."

Corrected: "... transcript sequences [24]. In a previous study [25], ..."

Line 91: "... transcriptome analysis[26,27] and ..."

Corrected: "... transcriptome analysis [26,27] and ..."

Line 94: "... across different tissues[28]."

Corrected "... across different tissues [28]."

Lines 113 to 125: "... 18S rDNA was amplified by polymerase chain reaction (PCR) using ITS1 (5'-TCCGTAGGTGAACCTGCGG-3') and ITS4 (5'-TCCTCCGCTTATTGATATGC-3') primers. ... phylogenetic tree based on the ITS region was constructed using MEGA5.1 software with the neighbor-joining method, and the statistical analysis utilized bootstrapping with 1,000 replications (Figure S1)."

Here, It is not clear what DNA fragment was used to construct the phylogenetic tree: The 18S rDNA, the fragment that was amplified (line 113) or a phylogenetic tree based on the ITS region (line 123). I suggest reviewing this item and explain clearly, what phylogenetic marker was used for phylogenetic analysis. I also recommend cite the reference of MEGA5 software.

Tamura K, Peterson D, Peterson N, Stecher G, Nei M, Kumar S. MEGA5: molecular evolutionary genetics analysis using maximum likelihood, evolutionary distance, and maximum parsimony methods. Mol Biol Evol. 2011 Oct;28(10):2731-9. doi: 10.1093/molbev/msr121. Epub 2011 May 4. PubMed PMID: 21546353; PubMed Central PMCID: PMC3203626.

Lines 119 to 120: "... Trans5 α Chemically Competent Cells(TransGen Biotech,Beijing,China) for DNA sequencing."

Corrected: "... Trans5 α Chemically Competent Cells (TransGen Biotech, Beijing, China) for DNA sequencing."

Lines 133 to 134: "... canned bottles(10 cm \times 15 cm, diameter \times height) with ..."

Corrected "... canned bottles (10 cm \times 15 cm, diameter \times height) with ..."

Line 141: "... body(Fig.2 F)."

Corrected "... body (Fig.2 F)."

Line 143: "... Waters e2695 system(Waters Technologies, Massachusetts,USA) ..."

Corrected "... Waters e2695 system (Waters Technologies, Massachusetts, USA) ..."

Line 145: "... Agricultural Industry Standard[29], ..."

Corrected "... Agricultural Industry Standard [29], ..."

Line 168: "... fragmentation buffer.Next ..."

Corrected "... fragmentation buffer. Next ..."

Line 176: "... using TBS380, ..."

I recommend describing the equipment TBS380: "using TBS380 Fluorometer (Turner Biosystems Inc., California, USA), ..."

Line 190: "... BLAST2GO ..."

I suggest citing the reference of BLAST2GO:

Conesa A, Götz S, García-Gómez JM, Terol J, Talón M, Robles M. Blast2GO: a universal tool for annotation, visualization and analysis in functional genomics research. *Bioinformatics*. 2005 Sep 15;21(18):3674-6. Epub 2005 Aug 4. PubMed PMID: 16081474.

Lines 199 to 200: "... fragments kilobase of exon per million mapped reads method."

I recommend writing the abbreviation of fragments per kilobase of exon per million mapped reads (FPKM).

Line 201: "RSEM(<http://deweylab.biostat.wisc.edu/rsem/>) [32] ..."

Corrected: "RSEM (<http://deweylab.biostat.wisc.edu/rsem/>) [32] ..."

The reference [32] is incorrect. The reference [32] is "Anti-cancer effects of cordycepin on oral squamous cell cancer..." The correct reference is the one numbered as [8] "RSEM: accurate transcript quantification from RNA-Seq data ..."

I suggest reviewing the citations and change the numbers and references appropriately.

Line 209: "0.05.In ..."

Corrected: "0.05. In ..."

Lines 213 to 215: I recommend citing the references of the bioinformatics tools used for bioinformatics analysis.

Klopfenstein DV, Zhang L, Pedersen BS, Ramírez F, Warwick Vesztrocy A, Naldi A, Mungall CJ, Yunes JM, Botvinnik O, Weigel M, Dampier W, Dessimoz C, Flick P, Tang H. GOATOOLS: A Python library for Gene Ontology analyses. *Sci Rep*. 2018 Jul 18;8(1):10872. doi: 10.1038/s41598-018-28948-z. PubMed PMID: 30022098; PubMed Central PMCID: PMC6052049.

Xie, C. et al. KOBAS 2.0: a web server for annotation and identification of enriched pathways and diseases. *Nucleic Acids Research* 39, W316–W322 (2011).

Wu, J., Mao, X., Cai, T., Luo, J., Wei, L. KOBAS server: a web-based platform for automated annotation and pathway identification. *Nucleic Acids Res* 34, W720–W724 (2006).

Line 225: "... previous researchesof ..."

Corrected: "... previous researches) of ..."

I also suggest citing the previous research.

Line 278: "... 3C).The identity ..."

Corrected: "... 3C). The identity ..."

Lines 294 to 295: "... and biogenesis," followed by "post-translational modifications ..."

Corrected: "... and biogenesis", followed by "post-translational modifications ..."

Line 313: "MAPK1_3(mitogen-activated ..."

Corrected: "MAPK1_3 (mitogen-activated ..."

Line 327: "... properties[34,35]."

Corrected: "... properties [34,35]."

Line 335: "...with a |log₂ ratio| ..."

I suggest describe that is the log₂ of the fold change: "... with a |log₂FC ratio| ..."

Line 336: "... less(Figure S4 ..."

Corrected: "... less (Figure S4 ..."

Line 343: "... comparisons(Figure 5B)."

Corrected: "... comparisons (Figure 5B)."

Line 369: "... analysis.Next, ..."

Corrected: "... analysis. Next, ..."

Lines 388 to 389: The authors described in the manuscript the analysis of the top 10 metabolic pathways and cited the Figure 3S B. However, the Figure 3S B is the top 20 metabolic pathways. I recommend reviewing this section.

Lines 388 to 389: The authors described in the manuscript the enrichment analysis and cited Figure S9. The authors detected that the most significant enriched pathway was biosynthesis of amino acids in the mycelium versus fruiting body... I suggest citing Figure S9 A, B and C as they appeared in this section. I also suggest marking with a red box or an arrow, for example, inside the figures (the most enriched pathway) for a better visualization.

Lines 419 to 425: I suggest using “Validation” instead “Verification” in 3.7 result topic (line 419). The authors the six targets used to the validation of DEGs by qRT-PCR and cited the Figure 1 (lines 424 to 425). Is it correct? Figure 1? Realtime results and comparison with RNA-Seq are in Figure 6. I recommend reviewing this paragraph.

Line 430: “... sclerotium .The expression ...”

Corrected: “...sclerotium. The expression ...”

Line 432: “... 5'-nucleotidase (c62060g1),surE (c19447g1), and CECR1(c34980g1)”

Corrected: “... 5'-nucleotidase (c62060g1), surE (c19447g1), and CECR1 (c34980g1)”

Line 446: “... previously published[17,19,36].”

Corrected: “... previously published [17,19,36].”

Line 451: “... play important roles in immuno-regulation of *C. cicadae*. .”

Corrected: “... play important roles in immuno-regulation of *C. cicadae*.”

Line 458: “... CECR1(adenosine deaminase) and surE(5'-nucleotidase) appeared ...”

Corrected: “... CECR1 (adenosine deaminase) and surE (5'-nucleotidase) appeared ...”

Line 564: “... deaminase(c35629g1, upregulated; c34980g1, downregulated) ...”

Corrected: “... deaminase (c35629g1, upregulated; c34980g1, downregulated) ...”

Line 466: “... substances[15], ...”

Corrected: “... substances [15], ...”

Line 467: “... biosynthesis[14]with ...”

Corrected: “... biosynthesis [14] with ...”

Line 471: “... deaminase,5'-Nucleotidase,Guanine deaminase, ...”

Corrected: “... deaminase, 5'-Nucleotidase, Guanine deaminase, ...”

Line 476: “... 5'-nucleotidase(c62060g1) gene.”

Corrected: “... 5'-nucleotidase (c62060g1) gene.”

Line 479: "... reductive mechanism[13]; ..."

Corrected: "... reductive mechanism [13]; ..."

Line 481: "... cordycepin.However, ..."

Corrected: "... cordycepin. However, ..."

Legend of Figure 6: I suggest reviewing the genes because there are two different genes with the same name CECR1 (B) and (D). I recommend writing the abbreviations together with the name. For example: (A) c19459g1 (ribonucleoside-diphosphate reductase subunit M1, RRM1). I also recommend writing the gene names in the Figure 6 below the graphics, next to the contig name: c19459g1 (RRM1).

Appendix B

Dear Editor:

We are truly grateful to yours and other reviewers' critical comments and thoughtful suggestions. Based on these comments and suggestions, we have made careful modifications on the original manuscript. All changes made to the text are in red color. In addition, we have consulted a native English speakers and Dr. Ying Huang for paper revision before the submission this time. We hope the new manuscript will meet your magazine's standard. Below you will find our point-by-point responses to the reviewers' comments/questions.

Response to reviewer 1 major comments:

After examining your comments carefully, we must admit that we have not expressed our meaning directly in the previous manuscript. In the revised version, we reorganized the means of expression, we integrated section 3.1,3.5 and 3.2 and marked red, and proposed our hypothesis of cordycepin biosynthesis pathway in *C.cicadae* in Figure 7. As suggested by the reviewer, we have invited an native English speaker to polished our manuscript.

Reponse to L15-24, Gene expression analysis in our manscript was Differential expression analysis, in order to identify differentially expressed genes(DGEs), and GO and KEGG analyses were aim to class the identified DGEs to metabolic pathways in *C.cicadae*. And Differential expression analysis has replaced in line 15. We conducted three comparisons, 1 mycelium (control) versus fruiting body, 2 mycelium (control) versus sclerotium, and 3 fruiting body (control) versus sclerotium, and they produced different numbers

of differential genes due to the cordycepin contents differed, highest (Table 2) in the sclerotium (embedded with sterile wet soil and total cultured for 29 days), then fruiting body samples, cordycepin was not detected in mycelium samples.

Response to Line 24 Why were the six genes chosen? In the literature of previously reported, the 5'-nucleotidase and adenosine deaminase has been proved to involved in cordycepin biosynthesis pathway, and RRM1 ,NDK were significantly up-regulated in mycelium (control) versus sclerotium.

Response to Line 66. Ophiocordyceps sinensis and Hirsutella sinensis are representing..... We are sorry for our negligence of this and thanks for your suggestions.

Response to Line 113-117 We use Taq PCR Mix to amplifying the ITS region, and the length of PCR products was 534 bp, PCR extension time of 90 s at 72°C is too long and in the future experiment we will decrease the extension time to 45 s, and we are sorry to Ignore the problem. Thanks for your kind remind.

Response L155-156 The cultured in canned bottles means after inoculated in Chinese Tussah silkworm pupae, they were cultured in an empty canned bottles Figure 2 A-D, and buried in soil means pupae were then embedded in canned bottles (10 cm×15 cm, diameter×134 height) with sterile wet soil and allowed to form sclerotium for 10 days with a 16:8-h. This was added in section 2.1 and made red.

Response to L238-250 This paragraph has been simplified and thanks for your kind advise.

Response to L253 What's the difference between a unigene and a transcript?

Because eukaryotes have variable shear, a gene may correspond to multiple transcripts; unigenes are the longest assembled transcripts and transcripts were also analyzed in our original research, unigenes have covered the analysis results of transcripts.

Response to L311, L314, and other parts.

All supplementary material sections have been replaced by “electronic supplementary material, name”. We are sorry for our negligence of this. We deleted the section 3.5 and insert into section 3.2.

Response to reviewer 1 minor comments:

Line 27 has been revised to “are responsible”.

Line 34 has added genus before *Cordyceps*.

Line 47 was also revised to *O. sinensis*.

Line 105-106 Stromata was the mat-like structure which formed by mycelium combines with host substrate. And fruiting body was a multicellular structure on which spore-producing structures.

L126 “strain buc” has been changed normal.

Figure 5B has corrected as MF, FS and MS.

Response to reviewer 2

Thank you very much for your kindly comments on our manuscript. Based on your suggestions, we carefully revised the manuscript.

- 1) *Cordyceps* in the text has been abbreviated;
- 2) *Ophiocordyceps sinensis* were more used than *Cordyceps sinensis* and we unified in *O. sinensis* in the whole manuscript.
- 3) I think the reviewer means Line 131 was line 141, and Fig. 2F has been replaced with Figure 2F

- 4) Line 331;the species names have be italicized in whole mancuript.
- 5) Line 347 GO has been revised in the title of section 3.2 .
- 6) Text in figure3-5 has been enlarged
- 7). The referenced format of Figures throughout whole manscript have been corrected.

Response to reviewer 3

We are truly grateful to yours critical comments and thoughtful suggestions. Based on these comments and suggestions.We conducted extensive proofreading, all citations and Figures has appiled with the Royal Society Open Science rules for authors. The table with annotation results is newly upload to figshare.

Line 38 has been corrected.

Lines 38 to 41: The references was cited here.

Lin53,55,56,61,64,70,77,83,85,86,87 **119-121,133-134,141,143,145,168,209,278, Lines 294 to 295,313,325,336,343,369** has all been corrected. And the reference of MEGA5 software was added.

line 125 ITS region was used for amplified and conduct phylogenetic analysis.

Line 176 has been replace with “using TBS380 Fluorometer (TurnerBiosystems Inc., California, USA)”.

Line 190, The reference of BLAST2GO was added.

Lines 199 to 200 The abbreviation of FPKM(fragments per kilobase of exon per million mapped reads) was added.

Line 201 The reference has been checked.

Line 213-215 The references of the bioinformatics tools used for bioinformatics analysis were cited.

Line 225 References 21-24 was cited in the previous research.

Line 388-389 The top 20 metabolic pathways was replaced. And Figure S9 ABC was cited in the text and a red box was marked in the figure.

Lines 419 to 425 Validation was replaced. The validation of DEGs by qRT-PCR and cited the Figure 6 (lines 424 to 425).

Legend of Figure 6 was revised as your suggesstions. The name of c35629 in Figure 6D was ADA2 and the gene names in the Figure 6 below the graphic were added.

Bsed on the comments we received, careful modifications have been made to the R1 manuscript. All changes were marked in red text. We hope that these revisions are satisfactory and that the revised version will be acceptable for publication in Royal Society Open Science.

Thank you very much for your work concerning my paper.

Wish you all the best!

Sincerely yours

Appendix C

Dear Editor:

We must thank you and all other reviewers for the critical feedback. We feel lucky that our manuscript went to these reviewers as the valuable comments from them not only helped us with the improvement of our manuscript, but suggested some neat ideas for future studies. Please do forward our heartfelt thanks to these experts.

Based on the comments we received, careful modifications have been made to the R1 the revised R2 version. We hope the new manuscript will meet your magazine's standard. Below you will find our point-by-point responses to the reviewers' comments/ questions:

Response to reviewer1

After examining the reviewer's comments carefully.

The title has been revised.

Introduction *Cordyceps cicadae* indeed can parasitize many hosts and we revised this confusion.

3.2 We delete the last sentence and this was indicated in 3.3 RNA extraction.

3.3 Because we select three comparisons due to the cordycepin contents, according to the results of HPLC determination. The cordycepin contents were highest (table 2) in the sclerotium (embedded with sterile wet soil and cultured for 29 days), followed by the fruiting bodies; cordycepin was not detected in mycelium samples.

4.1 Mycelium-1-3 has been replaced.

Literatures All authors were listed consistent with papers for literature 8,16,17, 19